# Aptasensor Integrated with Two-Dimensional Nanomaterial for Selective and Sensitive Electrochemical Detection of Ketamine Drug

**DOI:** 10.3390/mi15030312

**Published:** 2024-02-24

**Authors:** Shariq Suleman, Nigar Anzar, Shikha Patil, Suhel Parvez, Manika Khanuja, Roberto Pilloton, Jagriti Narang

**Affiliations:** 1Department of Biotechnology, School of Chemical and Life Science, Jamia Hamdard, New Delhi 110062, India; shariqsuleman07@gmail.com (S.S.); nigarsheikh111@gmail.com (N.A.); shikhapatil29@gmail.com (S.P.); zoyashadan414@gmail.com (S.); 2Department of Toxicology, School of Chemical and Life Science, Jamia Hamdard, New Delhi 110062, India; sparvez@jamiahamdard.ac.in; 3Centre for Nanoscience and Nanotechnology, Jamia Millia Islamia, New Delhi 110062, India; manikakhanuja@gmail.com; 4Institute of Crystallography of National Research Council (IC-CNR), Monterotondo, I-00015 Rome, Italy

**Keywords:** ketamine, diagnosis, paper electrode, recreational drug, electrochemical, sensors

## Abstract

Ketamine is one of the most commonly abused drugs globally, posing a severe risk to social stability and human health, not only it is being used for recreational purposes, but this tasteless, odourless, and colourless drug also facilitates sexual assaults when it is mixed with drinks. Ketamine abuse is a threat for safety, and this misuse is one of the main uses of the drug. The crucial role of ketamine detection is evident in its contributions to forensic investigations, law enforcement, drug control, workplace integrity, and public health. Electrochemical sensors have gained considerable interest among researchers due to their various advantages, such as low cost and specificity, and particularly screen-printed paper-based electrode (SPBE) biosensors have gained attention. Here, we reported an ePAD (electrochemical paper-based analytical device) for detecting the recreational drug ketamine. The advantages of using a paper-based electrode are that it reduces the electrode’s production costs and is disposable and environmentally friendly. At the same time, nanographite sheets (NGSs) assisted in amplifying the signals generated in the cyclic voltammetry system when ketamine was present. This ePAD was developed by immobilizing a ketamine aptamer on NGS electrodes. The characterization of proper synthesized NGSs was performed by Scanning Electron Microscopy (SEM), XRD (X-ray Diffraction), Fourier-transform infrared spectroscopy (FTIR), and UV-Vis spectroscopy. Electrochemical techniques, including cyclic voltammetry (CV) and linear sweep voltammetry (LSV), were employed to validate the results and confirm each attachment. Furthermore, the versatility of the proposed sensor was explored in both alcoholic and non-alcoholic beverages. The developed sensor showed a low LOD of about 0.01 μg/mL, and the linear range was between 0.01 and 5 μg/mL. This approach offers a valid diagnostic technique for onsite service with minimal resources. This cost effective and portable platform offers desirable characteristics like sensitivity and selectivity and can also be used for POC (point of care) testing to help in the quick identification of suspicious samples and for testing at trafficking sites, amusement parks, and by the side of the road.

## 1. Introduction

Ketamine (KT) (2-(2-chlorophenyl)-2-(methylaminocyclohexan), shown in Figure 1, is a common medical substance used for the induction and maintenance of anaesthesia [1,2]. KT is a hallucinogenic, anaesthetic, tasteless, and colourless drug that is widely used by both humans and animals [3]. However, ketamine abuse has increased in recent years. Ketamine is one of the drugs that can be identified either alone or in a mixture with other illicit narcotics, according to inspections in several nations [4]. Furthermore, the use of ketamine for recreational purposes, particularly by young people, has raised serious concerns [5]. Additionally, because it has no taste, colour, or odour, this makes it easier for sexual assaults when it is added to drinks or illicit drug formulas [6]. Due to such reasons, ketamine is now banned in most countries. The need for testing at locations of trafficking and amusement parks, as well as roadside testing, is increasing due to the requirement for speedy identification of suspicious samples. Presently, various methods such as GC-MS (gas chromatography–mass spectrometry), LC-MS (liquid chromatography–mass spectrometry), (SPE-LC-MS) solid-phase extraction–liquid chromatography–mass spectrometry, capillary electrophoresis, and other electrochemical methods are employed for ketamine detection in diverse samples [7,8,9,10]. However, these techniques still have certain drawbacks, such as the need for expensive equipment and specialized personnel, their time consuming sample preparation, and high costs, which restrict their broad use. So, a suitable method for rapid and efficient detection is urgently needed. In some cases, electrochemical paper-based sensors may replace these conventional techniques and avoid their limitations. Here, we examine nanographite with ketamine aptamer, which has shown potential for detecting recreational drugs electrochemically, and explain recent advancements in those sensors. Graphene is a 2D honeycomb structure made up of carbon atoms arranged in a hybridized honeycomb sp2 network, and it has been gaining popularity in recent years due to its unique characteristics, including ultrathin structure, high electrical conductivity, large surface area, and high theoretical capacitance [11,12,13,14]. These materials have been used in nanodevices, biomedical applications, biosensors, energy materials, etc. [15,16]. Moreover, NGSs have been selected as the preferred electrode material for supercapacitors due to their exceptional performance, longer cyclic stability, and cost-effectiveness compared to graphene [17]. The structure of NGSs boasts a unique porosity, facilitating convenient access for electrolyte ions. This feature makes them highly advantageous for energy storage purposes. While graphene and porous carbon materials are also widely used as electrodes in supercapacitors, NGSs have been found to have better cyclic stability and performance [18]. By utilizing functionalized nanographite as a versatile base, biomolecules can be easily immobilized, allowing for the sensitive detection of analytes within biological systems. Nanographites have a large surface area and ripples and are hydrophobic, which makes them appropriate for biomedical applications [19]. A nanographite is a type of graphene derivative that has varying oxygen, carbon, and hydrogen ratios. Another notable advantage of NGSs is their superior solubility, often surpassing that of graphene under suitable experimental conditions [20]. Nanographite possesses a negative charge, enabling it to engage with electroactive biomolecules through electrostatic interactions. The interaction between nanographite and its surroundings is heavily influenced by its electronic properties. These include its response to electric fields, energy dispersion patterns, and band gap [21]. Nanographite has been extensively utilized in the development of various electrochemical sensor types. In electrochemical sensors, the electrode functions as the conductor, representing a crucial type within chemical sensors. Chemically modified electrodes (CMEs) stand out as a modern electrochemical sensor type, achieved by introducing reagents to the electrode surface or within the electrodes to enhance their behaviour. A pivotal feature of CMEs lies in their ability to facilitate the oxidation or reduction of analytes, thereby reducing the high overvoltage required for voltametric measurements [22]. Aptamers serve as innovative recognition elements owing to their advantages over antibodies, which include strong affinity, low dimensionality, and exceptional stability. Additionally, they offer the benefits of easy alteration, in vitro synthesis, and being easy to design into sequences.

When a target binds, the electrochemical aptamer-based (EAB) sensor platform induces the folding of DNA and RNA aptamers, making it a frequently employed tool for detecting small-molecule targets such as ketamine [23]. The electrochemical aptamer-based (EAB) sensor is based on the target-binding-induced conformational change. The EAB sensor uses an aptamer that has been changed by a redox reporter and is electrode bound as the recognition element. Due to the conformational changes of the aptamer caused by target binding, the electron exchange rate between the redox reporter and electrode is altered. This can be easily detected as electrochemical output that can be used to monitor target concentration [24]. Ketamine is a drug that is electrochemically active and may transport electrons to the surface of an electrode at positive potentials [25,26,27]. Electrochemical sensing approaches are effective in identifying a range of electroactive compounds. They are simple, affordable, and only need a brief amount of analytical time as no time-consuming derivatization or extraction activities are necessary. They also offer information on the kinetics and mechanism of charge transfer in electrode processes. NGSs were used to modify the working electrode of developed ePADs. The issues with the traditional electrodes (carbon paste, glassy carbon, and graphite) often revolve around electrode surface fouling and excessively positive drug oxidation potential with interference from the oxygen evolution current. Some electroanalytical methods for determining ketamine have been published, although the majority of them are based on colorimetric and electrochemiluminescence principles [28,29,30]. Thus, electrochemical paper-based analytical devices (ePADs) have been developed to avoid difficulties with traditional electrodes. Furthermore, an aptamer for ketamine was immobilized on the surface-modified working electrodes, and ketamine was detected with the help of a ketamine aptamer and the results were validated via electrochemical techniques such as cyclic voltammetry (CV) and linear sweep voltammetry (LSV), which confirmed their respective attachments [31]. This study presents the analytical performance of a paper-based aptasensor incorporating NGSs and aptamers for the detection of the illicit drug ketamine in beverages. The proposed biosensor demonstrates promise as a sensitive and accurate diagnostic tool for ketamine detection. The novelty of our aptasensor lies in incorporating nanographite sheets into the three-electrode system, thereby improving the electrochemical performance of the sensor. The exceptional surface area and conductivity of nanographite-enhanced electron-transfer kinetics resulted in increased sensitivity and a significantly lower detection limit for ketamine. This paper presents an original paper-based three-electrode aptasensor, created in-house, designed for the swift and specific detection of ketamine, a powerful psychotropic substance, particularly in beverage samples. Paper-based electrodes show great promise as a cost-effective, portable, and user-friendly platform for developing sensors capable of detecting ketamine and other analytes. Moreover, in the future, these paper-based electrodes can be seamlessly integrated into portable, handheld, and wearable electronic devices for convenient readout of the sensor response. 

## 2. Material and Methodology 

### 2.1. Chemicals, Reagents, and Apparatus

Methylene blue (Purity: 82% and CN: 52015) was purchased from LOBA CHEMIE Pvt. Ltd., in Mumbai (India). Black carbon and silver conductive ink were ordered from Snab Graphix Pvt Ltd., Bangalore, India. For the synthesising nanographite, graphite synthetic powder, sulphuric acid, potassium permanganate, hydrogen peroxide, and hydrochloric acid were used. Graphite synthetic powder (purity: 98% and CN: 59538) was purchased from SRL (Sisco Research Laboratories Pvt. Ltd., Maharashtra, India). Ketamine (CN: FN04142107 and purity: 1.0 mg/mL in 1 mL distilled water) was purchased from Mtor Life Science Pvt Ltd. (New Delhi, India).

Aptamer Sequence:

(5′-GGG GGG ACG GGG CGG GAC GTG GTG TGT GGT TCG TGT CCC C-3′) [32]

#### 2.1.1. Apparatus/Instrument Used

Electrochemical measurements including CV and LSV were performed using Metrohm Dropsens (Stat-I 400s). The CV and LSV electrochemical experiments were conducted in the potential range between −1 V and +1 V at a scan rate of 50 mVs^−1^. Methylene blue (120 μL) was used as a redox probe with a pH of 7.4. Quanta 3D “FEG-FEI” technology was used to study the material’s surface shape using a field exhaust scanning electron microscope (FESEM). ‘Rigaku Smart’ Cu K X-ray (1.540 Å) was utilized to analyse the crystalline nature of the nanoparticles using X-ray Diffraction (XRD). The Agilent Cary100 series is UV-Vis spectrometer that was used to measure UV-Vis absorbance. Fourier-transform infrared spectroscopy (FTIR) was carried out employing a Bruker Tensor 37.

#### 2.1.2. Spiked Beverages: Alcoholic and Non-Alcoholic Drinks Were Taken for Spike Testing

To investigate the analytical approach of the sensor, various drinks were used, including alcoholic beverages (beer) and non-alcoholic drinks (coke and frooti), which were sourced from local stores in the market. The experiments on spiked drinks demonstrated the performance of our device for real sample detection in a complex environment containing various other molecules. It has been reported that illicit drugs, such as ketamine, are commonly spiked in drinks for recreational purposes and sexual assaults in various places, including clubs and restaurants. Therefore, detecting ketamine in both alcoholic and non-alcoholic drinks is crucial. 

### 2.2. Preparation of Standard Solutions

The ketamine-binding aptamer was prepared by mixing 221 µL of sterile distilled water in the main vial to make it 100 µM. For further dilution, 10 µL of aptamer was added to 190 µL of sterile distilled water. The final diluted form, i.e., 5 µM, was used for the experiment.

### 2.3. Synthesis of Nanomaterials 

#### Preparation of Nanographite (NG) Sheets

For the synthesis of nanographite (NG) sheets, 5 g of natural graphite powder and 100 mL of cooled, concentrated sulfuric acid (H_2_SO_4_) were combined, and the mixture was homogenized. Nitric acid was slowly added to the mixture while stirring. Potassium permanganate (KMnO_4_) in the amount of 12 g was then gradually added with ongoing agitation. The suspension was agitated for the next three hours at room temperature. Following that, 150 mL of double-distilled water (ddH_2_O) was added to the reaction, and it was stirred for 1.5 h. Afterward, 75 mL of 30% hydrogen–peroxide (H_2_O_2_) was added to the mixture. To maintain a pH of 7.4, the product was filtered, immediately washed with hydrochloric acid (HCl), and then rinsed with deionized water. To form a brown dispersion, the solution was diluted with double-distilled water. One additional round of filtration was performed on the product before being dried at 80° Celsius overnight to produce powdered graphite oxide. Graphite oxide (0.5 mg) powder was mixed with deionized water (10 mL). The above suspension was then sonicated for 30 min, resulting in the production of nanographite sheets [33]. 

### 2.4. Fabrication of Paper-Based Three-Electrode System 

Paper-based electrodes were constructed using a silk-screen frame and carbon conductive ink on paper. Carbon conductive ink was squeezed (hand printing) onto cellulose sheets using a squeezer through the defined overhead screen’s open portions. The electrode was created using a silkscreen as a stencil, and its proportions were then fixed and framed. The printed electrodes comprised of three electrodes: a working electrode (WE), counter electrode (CE), and reference electrode (RE) drop casted with Ag/AgCl. (As shown in Figure 2). Consequently, the three-electrode system-based biosensor was developed. In this study, the sensor is inexpensive, requires a small sample volume, and is easy to use due to the use of paper as a sensing surface, which makes this technique a good option for use in developing countries. Additionally, carbon conductive ink also provides beneficial properties including low cost, simple preparation, and speedy manufacture.

### 2.5. Preparation of Spiked Beverages

For the spiked study, a non-alcoholic drink was purchased from a regular retailer. The volume of each drink was 30 mL, respectively. Drinks were diluted in the ratio 1:1 in deionized water to obtain better results. Furthermore, drinks were spiked with ketamine and were simultaneously studied.

### 2.6. Immobilization and Deposition of Nanographite Sheets and Aptamer on a Paper-Based Sensor

On a working electrode of the paper-based sensor, 30 µL nanographite sheets were placed Further, the sensor was dried on a hot plate set at 60 °C for about an hour. The next step involved immobilizing 20 µL of aptamer on a working electrode embellished with nanographite sheets for about an hour at room temperature. Due to the small surface of working area, the aptamer was immobilised on the working region itself directly and was dried for 1 day. The aptamer was immobilised through the physisorption method. This modified sensor electrode enabled the detection of ketamine (volume 20 µL). Various concentrations (0.01 to 5 μg/mL) of ketamine were dropped along with MB 120 μL over this sensor and the hybridization was detected using CV and LSV. The sensor was used for alcoholic and non-alcoholic drinks where many interferent molecules were present, but the sensor showed insignificant interference. This confirms that the aptamer and NGSs are uniformly distributed onto the small working region of the miniaturized electrode.

### 2.7. Stages for Electrochemical Detection

In order to check the proper deposition of the nanomaterial and aptamer on the sensor, different stages of electrochemical analysis were performed. For this, the CV and LSV values of the electrode that had no deposited material (bare electrode) were examined. Following the overnight drying of the NGSs on the paper-based sensor, both voltammetry tests (CV/LSV) were performed as usual. Consecutively, 20 μL aptamer was then applied to the paper-based sensor that contained dried NGSs; after that, CV and LSV measurements were recorded. The last step was depositing the analyte molecule ketamine onto electrodes that contained both the aptamer and NGSs. Methylene blue (120 μL) was then used as a redox probe to conduct the CV/LSV and the results were analysed.

### 2.8. Optimization of Different Parameters and Investigation of Repeatability and Stability and Methods for Real-Sample Analysis of Spiked Beverages

Various concentrations of ketamine, ranging from 0.01 µg/mL to 5 µg/mL, were immobilized across various electrodes (which had already been modified with NGSs and aptamers) and then were held for drying. Following this, the response for ketamine detection was observed at each concentration using CV and LSV. The aptamer-immobilized electrodes were employed to detect ketamine at various temperatures (15, 25, 35, 45, 55, 65 °C) and times (5 to 35 s), and the best cyclic response was observed. The target was detected using the electrodes that had been immobilized with the aptamer. To corroborate the results, an electrochemical study was carried out after adding the target ketamine and MB to the electrode. 

By incorporating a predetermined amount of ketamine into a spiked beverage, the sensors’ capacity to function in real samples was evaluated (alcohol, coke, and frooti). A paper-based device was treated with this solution and the hybridization indicator, methylene blue. Electrochemical assessments were carried out to verify the findings. Ketamine concentrations were repeatedly measured, proving the repeatability of the suggested biosensors, and its stability was confirmed for at least a month.

## 3. Results and Discussion

### 3.1. Sensing Strategy and Acquiring Signals

According to the electrochemical method, which is based on a compound’s distinctive electrochemical profile, the target substance, ketamine, is an electroactive compound. These electroactive drugs involve the oxidation of amino groups and an aromatic nucleus [34,35]. Methylene blue acts as a redox mediator in electrochemical ketamine sensors, facilitating the transfer of electrons between ketamine molecules and the sensor electrode. This electron transfer results in a measurable signal that can be used for the sensitive and selective detection of ketamine. Following the introduction of the target, i.e., ketamine, with an aptamer forming a loop and associating with the target, the current response increases due to the electroactive behaviour of ketamine. As the ketamine concentration increases, the current response of the ket/aptamer/NGSs also increases. This is probably due to ketamine’s specific affinity for the guanine base rather than other aptamer sequence bases. Therefore, the interaction between aptamer and ketamine showed a dependency on the types of bases [36].

Ketamine has been identified by using screen-printed electrodes (SPEs) at a low cost. The huge surface area and rapid electron transfer kinetics of the NGSs are the fundamental drivers of the enhanced current in the circular region of the ePAD integrated with them (as shown in Figure 3). When voltage is applied to the sensor surface of the modified ePAD combined with NGSs and an aptamer, ketamine is oxidized.

### 3.2. Characterization of Nanographite Sheets

The following characterization procedures were used to verify that the synthesized nanographite sheets were successful: UV-vis spectrometry, FESEM, XRD, and FTIR. By using FESEM micrographs, the morphological confirmation of nanographite sheets was investigated. Figure 4a indicates that the NGSs has wrinkles and folds, ripple surfaces, consistent size particles, and a thin coating of an aggregate that is randomly organized [37]. An analysis of the FESEM image at a 300 nm scale enabled a precise determination of the thickness of the NGSs. Employing image analysis software, we systematically examined individual nanosheets, applying stringent selection criteria to ensure the precision of thickness measurements. The resulting thickness of the NGSs was approximately 25 nm [38]. Figure 4b illustrates the NGSs’ X-ray Diffraction pattern. The NG crystalline properties and interlayer changes were examined using XRD. The sharp peak at about 2θ = 10°corresponding to the (001) plane indicated the typical peak of the NGSs (JCPDS card no. 75-2078) [39,40]. The peak at an angle 2θ = 25.4° (002), which was shallow, is assigned to graphite, which was shifted to 10° after oxidation of graphite to NG. A broad peak at a range of 11° to 25° might be a sign of incomplete oxidation of graphene oxide, which forms a nanographite [41]. The severe oxidation method utilized to prepare the graphite powder and the intercalation of water molecules caused the oxygenated functional group on NG to develop a considerable interlayer gap. This outcome was initially proved for the proper formation of NG. Figure 4c displays the UV-Vis spectra of the synthesized NGSs. The aromatic C-C bond, π-π* transition, caused the UV absorption peak of the NGSs to occur at 240 nm [42]. As shown in Figure 4d, the FTIR spectroscopy was used to determine the functional groups and bonds present in the nanographite sheets. The oxygen moieties in the structure of NG include the vibration modes of epoxide (C–O–C) (1373 cm^−1^) and sp^2^ -hybridized C=C (1625 cm^−1^ in-plane vibrations, respectively). Additionally, the carboxyl (COOH) groups were identified through vibrations within the range of 1650–1750 cm^−1^ and the presence of C–OH vibrations at 3163 and 1043 cm^−1^ [43,44].

### 3.3. Electro-Chemical Properties of Ketamine/Aptamer/NGSs at Different Stages

By using the electrochemical CV and LSV technique, the electrochemical characterization of the ketamine/aptamer/NGSs paper-based modified electrodes was performed successfully. The differential current response has been verified by both CV and LSV at various electrode stages, as shown in Figure 5a,b. The diminished peak current response observed in CV for bare electrodes is attributed to slower electron transfer kinetics. Conversely, upon deposition of NGSs on the working electrode surface, a significant two-fold increase in current responsiveness is evident due to the rapid electron transfer kinetics of the NGSs. Because the aptamer (biological recognition element) is non-conductive, the current is significantly reduced once the aptamer is immobilized onto the working surface area. After the addition of a target, i.e., ketamine, with methylene blue, the current response gets increased due to the electroactive behaviour of ketamine. The same goes for the response in LSV. 

In LSV, bare electrodes exhibit a smaller peak current response due to slower electron transfer kinetics, whereas the deposition of NGSs leads to a noteworthy two-fold increase in current responsiveness. The immobilization of the non-conductive aptamer on the working surface further reduces the current. 

### 3.4. Effects of Different Ketamine Concentrations on the Aptamer/NGSs Paper-Based Sensor

To illustrate the designed sensor’s performance in terms of quantitative measurements, various ketamine concentrations were examined. Different concentrations of 0.01, 0.1, 1, 3, and 5 µg/mL were employed for the aptamer hybridization. The results specified that ketamine exhibits crossing with the aptamer, and different concentrations yielded varying current responses, thereby confirming the quantitative presentation of the sensor. These findings align with previously reported sensors. The observed increase in current response with rising ketamine concentration is attributed to the compound’s electroactive nature. It was observed that the detection limit was 0.01 µg/mL. Using two electrochemical measures, namely CV and LSV, the concentration findings were confirmed (As shown in Figure 6a,b). The peak current values and log of ketamine concentration were found to have an interesting linear association (as shown in Figure 6c,d). 

### 3.5. Optimization of Ketamine/Aptamer/NGSs Paper-Based Sensor in Terms of Temperature and Time

An important step in enhancing the performance of the developed sensor involves optimizing the biosensor. Temperature and time have an impact on the performance of the sensor. To achieve maximum responsiveness, in terms of these experimental settings, the sensor was updated. The developed ketamine/aptamer/NGSs paper-based sensor’s performance was investigated thoroughly in a range of temperature and time conditions. Cyclic voltammetry of the paper-based ketamine/aptamer/NGSs sensor was recorded at scan rates of 50 mV/s^−1^, spanning temperatures from 15 °C to 65 °C. The best current response was observed at 35 °C. Therefore, the sensor was optimised at 35 °C because the results were satisfactory. Notably, subsequently, the sensor’s optimization was explored at various time intervals (5 to 35 s). It was observed that the current increased from 5 s to 35 s, with the optimal current response time identified between 20 and 30 s. As a result, a deposition time of 25 s was chosen for further investigation. This finding is illustrated in Figure 7a,b. Consequently, the sensor was optimized for enhanced performance within the specified temperature and time parameters.

### 3.6. Limit of Detection and Accuracy (Recovery) Test

The limit of detection (LOD) is defined as the lowest target concentration at which the electrode can reliably detect the target. Here, the LOD was determined to be 0.01 µg/mL. To assess the biosensor’s recovery capability, 0.01 µg/mL of ketamine was introduced into various ketamine concentrations. The same process was repeated with other concentrations as well. The observed current closely approximated that of 0.1 µg/mL, resulting in a calculated recovery percentage of 97% and 95%. The accuracy was further evaluated through a recovery test and the results are given in Table 1. In the spike recovery method, a known concentration of the analyte (ketamine) was injected into the sensing solution and its concentration was measured by the sensor, taking the calibration curve as the standard. Mathematically, this can be explained by the equation
% Recovery = Observed/Expected × 100
where Observed is the value of analyte concentration observed after the spiking of the analyte; 0.01 μg/mL is the value of analyte concentration present in the sample before the spiking; and Expected is the exact concentration of the analyte spiked in the sample.

### 3.7. Examination of Cross-Reactivity (Specificity/Reliability) and Stability

Ketamine (0.01 µg/mL) was used to investigate the cross-reactivity performance of the paper-based sensor. The sample’s current response with CV is shown in Figure 8a,b, which demonstrate that the peak flow of the drug methamphetamine (0.01 µg/mL) is nearly equal to the aptamer/NGSs/ePAD, although the current rises when it is compared to ketamine. Also, the paper-based sensor was kept at 4 °C for different lengths of time (1 day, 7 days, 15 days, and 30 days) to detect ketamine (0.01 µg/mL) and its stability was evaluated using the CV approach. By the 15^th^ day, the created sensor’s findings were nearly on par with those of the aptamer/NGSs paper-based sensor, as shown in Figure 8c. The stability graph also displays the error bar, which provides enough repetition and reliability for ketamine identification.

### 3.8. Analysis on Spiked Beverages

The feasibility of the proposed method for detecting ketamine in samples (spiked beverages) was investigated using a conventional addition method to validate the usability of the built paper-based sensor. In particular, 0.01 µg/mL ketamine in beverages was spiked to the surface of the aptamer/NGSs ePAD, and peak current CV value testing was conducted, as seen in Figure 9a. In samples, the sensor responded satisfactorily, and the reaction was remarkably like ketamine. In order to evaluate the repeatability of our findings, multiple independent measurements (*n* = 5) were conducted for each spiked beverage. The developed sensor was able to detect ketamine in different beverages (alcohol, coke, and frooti) since the current response was found to be almost like ketamine alone. Real samples (alcohol, coke, and frooti) can include a wide variety of substances and impurities. The reduction signal may be more affected by some solvents or components of the matrix than the oxidation signal. By concentrating on the oxidation signal, these components’ interference can be lessened. A bar graph with error bars (*n* = 5) showing the current study of different spiked beverages (alcohol, coke, and frooti), compared with the other ePAD, is shown in Figure 9b. The current approach showed lower ketamine detection and linear range when compared to the previously reported study as shown in Table 2.

## 4. Conclusions and Future Prospective

Ketamine abuse is becoming more common, to the point where it is a serious concern. Electrochemical sensors have attracted a lot of interest because of their critical role in the early detection of illicit drugs in drinks, as well as other fluids. The electrochemical sensor has provided a variety of unique characteristics like better specificity, sensitivity, reproducibility and stability, low costs, increased surface to volume ratio, better electron kinetics, and fast reactions. Here, we rationally exploited the exceptionally high charge transfer efficiency of NGSs to develop a detection platform for ketamine, a recreational drug. Using an ePAD enhances the sensor even more because paper is a cheap substrate that can be produced in large quantities. Paper-based testing offers a point-of-care diagnostics platform that is affordable. These sensors are known as eco-designed analytical techniques due to their ecofriendly substrate, efficient production, and capacity to reduce waste management by incinerating the sensor after use. The LOD of the reported sensor was about 0.01 μg/mL and the linear range was between 0.01 and 5 μg/mL. It is obvious that the suggested sensor requires significantly less time and money than current analytical techniques for ketamine detection. XRD, SEM, and UV-vis spectroscopy were used to characterize the synthesized nanographite sheets. Aptamers were used in this study because they are unique and highly sensitive tools for use in rapid diagnostic approaches. CV and LSV, the two methods used to measure the analytical response of the biosensor, were validated using the potentiostat. The possibility of monitoring ketamine for pharmacokinetic and pharmacodynamic purposes, as well as accurate measurement in spiked samples, might become possible with this easy, quick, and inexpensive sensor. Based on the findings of this investigation, the developed ePAD might be used as a sensitive electrochemical-based sensor for drug analysis in the future.

## Figures and Tables

**Figure 1 micromachines-15-00312-f001:**
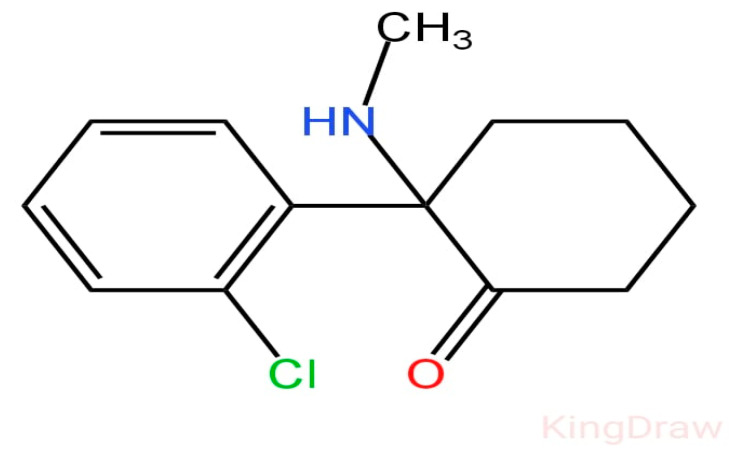
Chemical structure of ketamine drug.

**Figure 2 micromachines-15-00312-f002:**
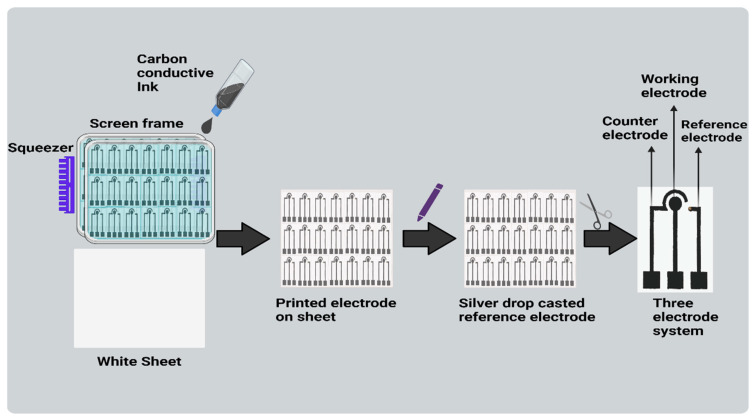
Fabrication is shown schematically of three-electrode system-based biosensor.

**Figure 3 micromachines-15-00312-f003:**
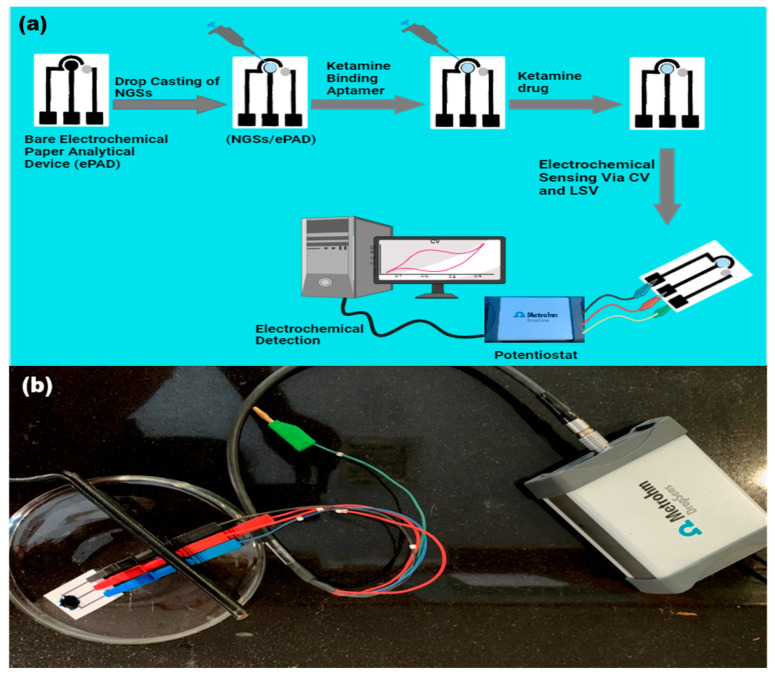
(**a**) Schematic representation of the fabrication and operation of three-electrode system. (**b**) The real image of final sensor.

**Figure 4 micromachines-15-00312-f004:**
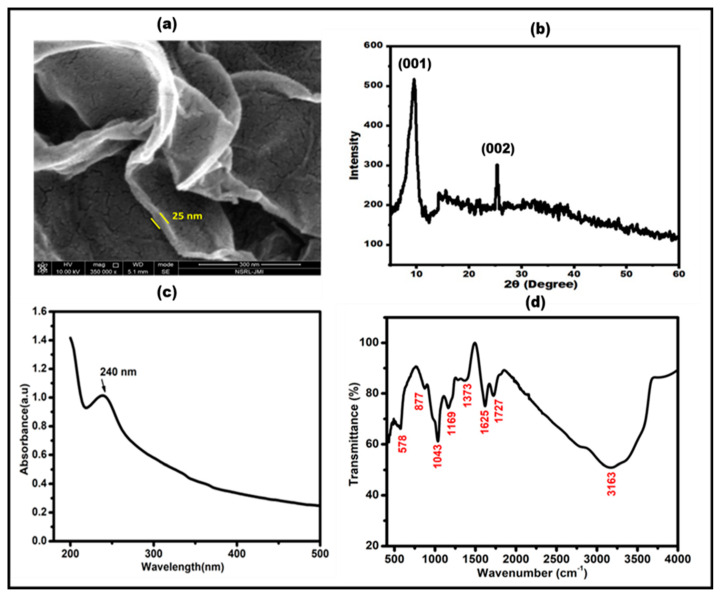
Characterization of nanographite sheets. (**a**) FESEM micrograph (**b**) X-ray Diffractogram (XRD), (**c**) UV-VIS absorption spectrum of NGSs, (**d**) FTIR spectrum of synthesized NGSs.

**Figure 5 micromachines-15-00312-f005:**
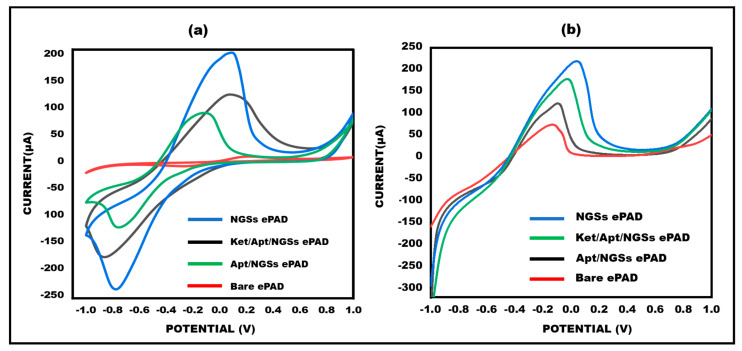
(**a**) A total of 10 mM methylene blue in 0.1 M KCl was subjected to cyclic voltammetry at bare ePAD, NGSs ePAD, aptamer/NGSs ePAD, and ket/aptamer/NGSs ePAD at 50 mVs^−1^. The potential range from −1 V to +1 V. (**b**) A total of 10 mM methylene blue in 0.1 M KCl was tested using linear sweep voltammetry in the range of −1 V to +1 V at 50 mVs^−1^ at bare ePAD, NGSs ePAD, aptamer/NGSs ePAD, and ket/aptamer/NGSs ePAD.

**Figure 6 micromachines-15-00312-f006:**
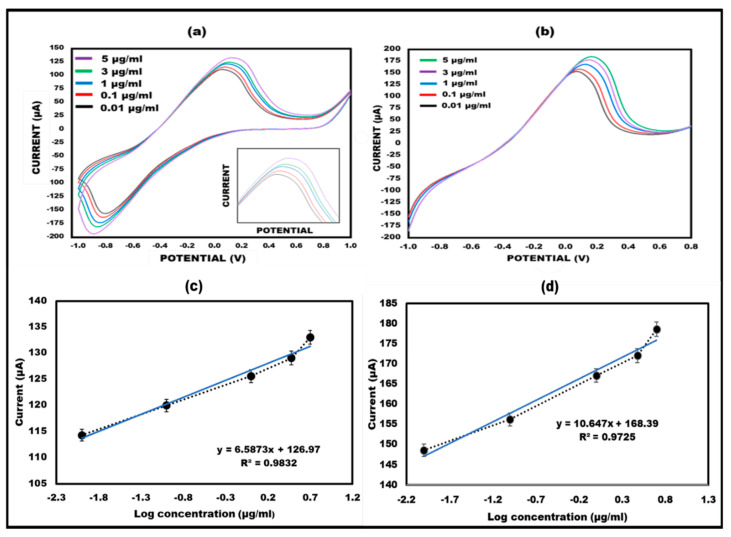
(**a**) CV of 10 mM methylene blue in 0.1 M KCl at ket/aptamer/ NGSs ePAD at 50 mVs^−1^ at varied concentrations ranging from 0.01 to 5 g/mL with the potential range from −1 V to + 1 V. (**b**) A total of 10 mM methylene blue in 0.1 M KCl was tested using linear sweep voltammetry at a ket/aptamer/ NGSs ePAD at 50 mVs^−1^ in the potential range of −1 V to + 1 V at various concentrations ranging from 0.01 to 5 µg/mL. (**c**) Linearity of CV with log concentration. (**d**) Linearity of LSV with log concentration.

**Figure 7 micromachines-15-00312-f007:**
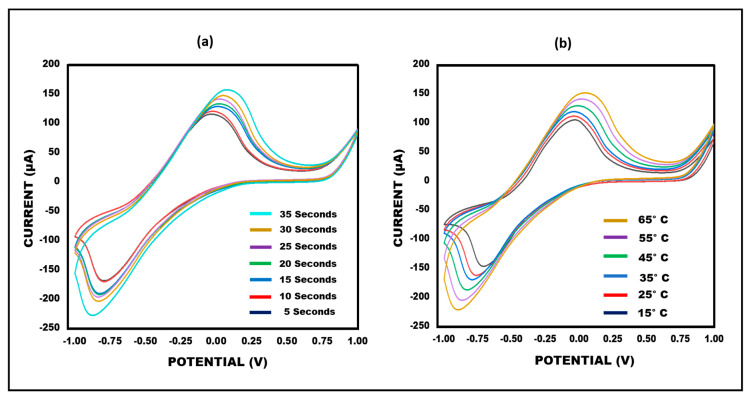
Cyclic voltammetry obtained for ket/aptamer/NGSs paper-based sensor for different (**a**) times (5 to 35 s) and (**b**) temperatures (15 to 65 °C) in 10 mM methylene blue in KCl at 50 mVs^−1^.

**Figure 8 micromachines-15-00312-f008:**
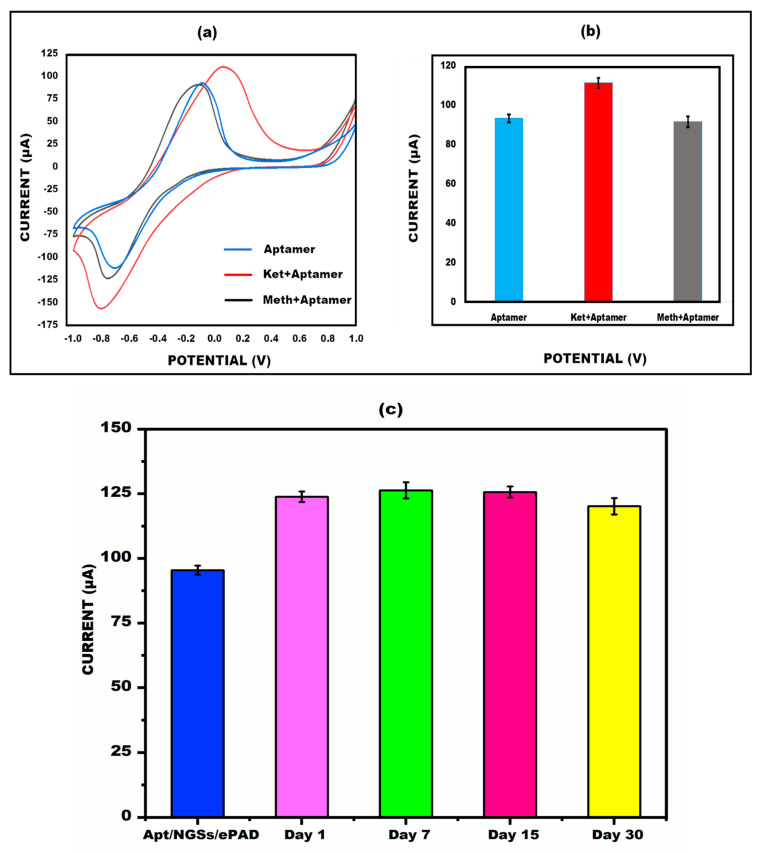
(**a**) Analysis of peak current cyclic voltammetry value of ketamine/aptamer/NGSs sensor binding with the interferent, methamphetamine. (**b**) Bar graph showing the comparison of (aptamer alone), (ketamine+aptamer), and (methamphetamine+aptamer). (**c**) The electrochemical test was used to evaluate the stability of the paper-based sensors capacity to detect ketamine on day 1, 7, 15, and 30, with error bars (*n* = 5).

**Figure 9 micromachines-15-00312-f009:**
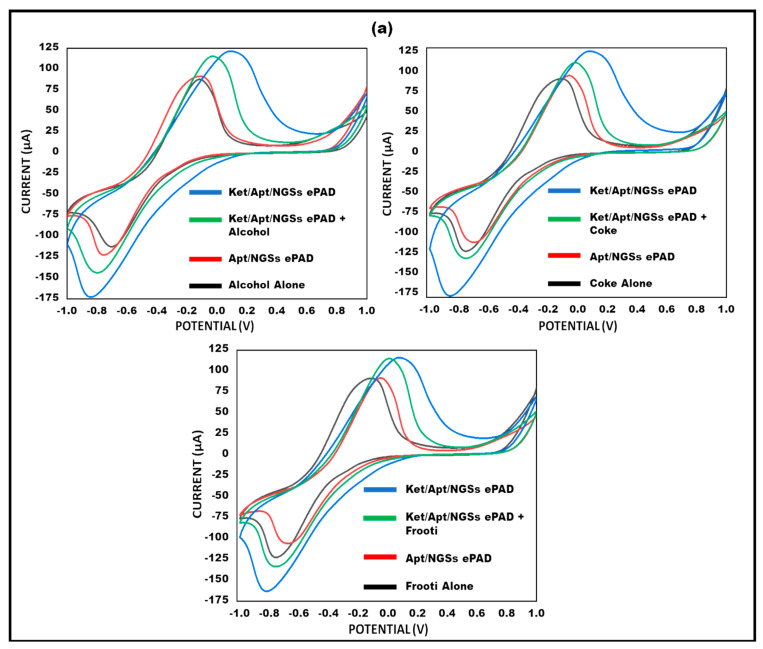
(**a**) Cyclic voltammetry peak, current study of ketamine on spiked beverages: (**a**) alcohol, Coke, frooti using fabricated paper-based sensor. (**b**) Bar graph with error bars (*n* = 5) showing current study of different spiked beverages compared with the other ePAD.

**Table 1 micromachines-15-00312-t001:** Recovery test of the built-in biosensor for ketamine.

Initial Concentration μg/mL	Concentration Added (µg/mL)	Current Measured (µA)	Expected Current Measured (μA)	Recovery (%)
0.01	0.1	112.651	115.608	97%
0.01	1	116.314	121.541	95%

**Table 2 micromachines-15-00312-t002:** Comparison of analytical performance of different ketamine sensors.

Methods	Linear Range (µM)	LOD (μM)	References
Electrochemical aptamer-based (EAB) sensor	0.01–3.0 μM	0.01 µM	[45]
Fluorimetry	20–1000 µM	200 µM	[46]
Colorimetry	105–315 µM	4.1 µg	[47]
Potentiometric sensor	9–10,000 µM	7.3 µM	[48]
Fluorescent carbon dots (CDs)	0.5–650 µM	230 nm	[49]
ePAD sensor	0.01–5.0 μM	0.01 µM	Proposed sensor

## Data Availability

Data is contained within the article.

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
