# Peer review of "Aptasensor Integrated with Two-Dimensional Nanomaterial for Selective and Sensitive Electrochemical Detection of Ketamine Drug"

_micromachines, 2024, doi:10.3390/mi15030312_

Round 1

Reviewer 1 Report (New Reviewer)

Comments and Suggestions for Authors

Figure 6: From the figure c and d, the linear response range is obviously between the first and the forth data points, as the last data points are off.

Comments on the Quality of English Language

Line 164: "The process used to synthesize graphene oxide nanosheets was cheap" This sentence does not describe the synthesis procedure and therefore should be removed.

Line 171: "HCL" should be HCl

Line 180: "Using a squeezer, carbon conductive ink was applied on cellulose
sheet. " The dangling modifier was not properly used here. Ink cannot use a squeezer.  Similar issues should be corrected throughout the paper.

Author Response

attached file

Reviewer 2 Report (New Reviewer)

Comments and Suggestions for Authors

In this manuscript, the author reported an ePAD (electrochemical paper-based analytical device) for detecting the recreational drug ketamine. Here, they rationally exploited the exceptionally high charge transfer efficiency of an GO NSs to develop a detection platform for ketamine- a recreational drug. This manuscript can be accepted after major revision.

1.       The position of the reference number varies across different locations. Some are inside sentences, while others are outside sentences. Please use the journal's reference format.

2.       Why did the author use Methylene Blue instead of K3[Fe(CN)6] and K4[Fe(CN)6] which are the ideal redox probes?

3.       During the detection of ketamine using CV and LSV (Figures 6a and b), the potential of the current response is not fixed. This is not acceptable. Otherwise, provide a rational explanation.

4.       Please mention the novelty issue clearly in the introduction section.

Comments on the Quality of English Language

No

Author Response

attached file

Round 2

Reviewer 1 Report (New Reviewer)

Comments and Suggestions for Authors

Comments on response 1: While R2 indeed indicate the goodness of the fit, itself cannot be used to determine a linear regression region. Instead, a significant decrease of R2 from incorporating more points should be used to exclude the points that are out of linear region.

Revised manuscript section 3.6, line 419-426.

The referee was confused about the recovery percentage calculation. In the text, "0.01 μg/mL of ketamine was introduced into various ketamine concentrations", while in the table, there is only one initial concentration, and the concentration of added ketamine becomes 0.1 ug/ml.

The authors should describe the method used here in details.

Figure 9: In Figure 9 (a), only one set of experiments were presented, and there is no description in experimental section that multiple measurements were performed. The referee assumed that the error bar in Figure 9(b) here was generated from error propagation. The referee was wondering if the author could show how these error bars were generated? 

Also, a legend or description in caption should be provided to link the data to its corresponding entry.

Table 2: What is the unit of linear range for Fluorimetry, Colorimetry, etc.?

Author Response

attached file

Reviewer 2 Report (New Reviewer)

Comments and Suggestions for Authors

I have no further comments. I recommend accepting the manuscript.

Author Response

attached file

Round 3

Reviewer 1 Report (New Reviewer)

Comments and Suggestions for Authors

 Accept in present form

This manuscript is a resubmission of an earlier submission. The following is a list of the peer review reports and author responses from that submission.

Round 1

Reviewer 1 Report

Comments and Suggestions for Authors

The manuscript presents the fabrication of ketamine sensor based on aptamer/GO sensing material. The topic is quite interesting to broad readership. However, many parts should be clarified/revised before publication in Micromachines.

- It’s not clear the role/benefits of GO and aptamer on ketamine sensing in this work. Aptamer was used for selectivity? Without aptamer, the sensors seem to be sensitive to ketamine as shown in Fig. 5. Please clarify. The pure aptamer sensor on  ketamine sensing should be also included in Fig. 5.

- The real photograph of finish sensors should be included.

- Please add the JCPDS Cards number for XRD. 

- Based on characterization of Graphene oxide, it’s not clear the quality of Graphene oxide. Normally, XPS/FTIR must be used to identify the functional groups and Raman/TEM should be used to see the layer, thickness, quality etc.

- A table should be provided for highlighting your work in ketamine sensing performance through making comparison with previous publications.

- It’s well known the pH strongly affects on chemical reaction for electrochemical sensors. The effects of pH should be investigated.

- The peak potentials are always shifted for all experiments referring to unstable chemical reactions. Authors must give a reason and explain more details for potential shift. 

Comments on the Quality of English Language

Moderate editing of English language required

Author Response

attached file

Reviewer 2 Report

Comments and Suggestions for Authors

The manuscript discusses the performance of aptasensors and the detection of the illicit drug ketamine using electrochemical techniques. Although the drug determination methods are very promising and important in many areas of expertise, interpretation of the electrochemical part of the work is far from ideal. I cannot recommend this manuscript for publishing in the current state. Authors should make some important changes before resubmission:

1)      Please check the spelling of the authors names.

2)      Thorough English editing through the manuscript is needed.

3)      Lines 15-17 and 30-32 in the Abstract duplicate each other. What electrochemical method was used for ketamine quantification? Add appropriate information to the abstract.

4)      There is poor information about aptamer immobilization. Authors did not mention modification of terminal aptamer groups (introduction of biotin, thiol groups, etc.). What was the nature of processes resulted in the aptamer /GO immobilization?

5)      In my opinion, misunderstanding of cyclic voltammetry (CV) and linear sweep voltammetry (LSV) results takes place in the manuscript. Interpretation of these experiments was very unexpected. In fact, principles of potential applying are the same in the CV and LSV modes. What was the sense of choosing two similar methods? CV registers both the reduction and oxidation branches of voltammograms, LSV – only one of them. In this connection, it seems unclear why the oxidation signals presented in Figures 5a and 5b have different dependences on the modifying layer content.

6)     It is not acceptable to use absolute current values instead of the peak currents in CV and LSV. Thus, all the experimental data should be recalculated and all dependencies should be plotted with new data.

7)       Even though absolute currents were used instead of the peak currents, other huge problems exist in calibration curves interpretations. Both the dependences presented were nonlinear. Only 4 points could be found within the linear range -  too little. Additional concentrations of ketamine should be measured and added to the concentration dependence.

8)      Authors did not use any buffer media for electrochemical investigations. The pH of 0.1 M KCl solution with 10 mM Methylene blue was not measured. Ketamine has an amino group in its molecule, so the pH value control is very important.

9)      If the optimal temperature was chosen as 35 ºC, how could the measurements be performed outside the lab? What is “intervals in seconds”? Is it ketamine incubation time?

10)  Authors cannot explain the Methylene blue role. It varies from “supportive electrolyte” to the "hybridization indicator". Both explanations are wrong. There is only one aptamer in the sensor assembly and no hybridization event occurred. Methylene blue interacts with aptamer and plays the role of redox probe, not of the supporting electrolyte.

11)  Authors cannot clearly explain the nature of electrochemical signal detection. Ketamine redox signal, Methylene blue signal and even guanine signal were mentioned through the manuscript text.

12)  Part 2.1 should be rewritten. Authors should avoid reagents listing as follows: “ For the fabricating electrodes:…”, “For the preparation:…”. Catalogue numbers of reagent, purity, manufacturers should be presented everywhere.

13)  Detailed description of CV and LSV electrochemical experiment such as potential ranges, scan rates, buffer solutions, pH should be presented in part 2.1.

14)  Detailed description of apparatus (manufacturer, country, city) should be presented in part 2.1.

15)  Line 133:” Spiked Beverages: Alcoholic & Non-Alcoholic drinks were taken for Spike Testing”. Add information about beverages sources.

16)  If 10 µL of 100 µM solution was mixed with 190 µL of sterile distilled water, resulting concentration should be 5 µM. Please check it.

17)  Please check indices in chemical formulas (for example, (H2SO4)) in Part 2.3.

18)  Please use Arabic numbers in the reference list.

19)  Lines 53-56:”Currently” was used twice in one sentence. Please check it.

20)  Check the phrases: Lines 75-76: ” Another advantage of GO is that, it frequently dissolves more easily or better solubility than graphene under the suitable experimental conditions.” Lines 80-82:”Modern-electrochemical sensors called chemically modified electrodes (CMEs) are constructed by applying reagent to the surface or within the surface of electrodes to improve their behaviour.”, Line 145: “The reaction was then given 150 mL of ddH2O (double distilled water), and it was stirred for one and a half hour.”, Line 317:” The developed-sensor ketamine/Aptamer/GO NSs paper-based sensor's performance was thoroughly investigated in a variety of temperature and time ranges.”

21)  Line 79:” An electrode serves as the conductor in electrochemical-sensors, which is an important type of chemical-sensors.” What electrode did the authors mean exactly?

22)  Line 83:” One of CMEs' most important features is their capability to catalyse the (oxidation/reduction) of analytes by reducing the high-overvoltage necessary for the voltametric measurement.”  - please use “facilitate” instead of “catalize”.

23)  Lines 85-87: Please divide the phrase presented in two sentences, add more information about the aptamers.

24)  Line 147: “To preserve pH, the product was filtered, immediately washed with hydrochloric acid (HCL), and then rinsed with deionized water.” What pH was discussed, acidic or neutral one?

25)  Line 151:”graphite oxide” or “graphene oxide”?

26)  Line 160: Please use “drop casted” instead of drop cast”.

27)  Line 162: Is it really possible to make reusable paper electrodes? What about the impregnation of the paper? There is no information about it in the manuscript.

28)  Line 170:” The concentration of each drink was 30 mL respectively” should be changed with “The volume of each drink was 30 mL respectively”.

29)  Please check the abbreviations: SPEs, SPBEs and ePAD – all of them were used in the manuscript.

30)  Parts 2.9 and 3.1. fully duplicate each other.

31)  Please explain the phrases below: Line 297: “The results showed that ketamine exhibits aptamer hybridization…”, Line 320:” At 65 degrees Celsius, the highest reaction was seen, although it was close to the aptamer.”

32)   Line 325:” between 30 and 35 seconds, it takes more time to display the response at the current which delays the sensors detection capability.” CV registration time depends on the scan rate and doesn’t depend on the incubation time. Please clarify it.

33)  Part 3.6.:” When 0.01 μg/mL of ketamine was added to other ketamine concentrations to show the suggested biosensor's excellent recovery, that triggered the current to be almost equal to 0.1 μg/mL and recovery percentage was calculated which was found to be 102%.” Summary concentration of  spiked ketamine was equal to 0.11 μg/mL. Why the recovery was calculated relative to the 0.01 μg/mL?

34)  Please add the methamphetamine concentration used in the experiments.

35)  Real samples (alcohol, coke) lead to significant decrease of sensor signal, especially at the reduction branch. Why was it not discussed in the manuscript? Why was only oxidation signal taken for the quantification?

36)  Article referenced as [37] in the Table 2 was absent in the List of References.

37)  There are huge amount of hyphen phrases that strictly hampers the manuscript reading: Lines 17 and 48:” sexual-assaults”, Line 45:”illicit-narcotics”, Line 46: “several-nations”, Line 49:”illicit-drugs”, Line 51:” amusement-parks”, Line 56: “ketamine-detection”, Line 58:” consuming-sample”, “high-cost”, Line 61:” avoid-their”, Line 62:” recreational-drugs”, Line 64:”honey-comb” or “honeycomb”, Line 79:” electrochemical-sensors”, Line 80: “chemical-sensors. Modern-electrochemical”, Line 89: “target-induced”, Line 92:” positive-potentials”, Line 99:” oxidation-potential”, Line 105:” surface-modified-working”, Line 108:” respective-attachments” , Line 110:” Drug-Ketamine”, Line 247:” graphene-oxide”, Line 266:” Electro-chemical”,  Line 175:”hot-plate”, Line 243:” Schematic-representation”, Line 317:” developed-sensor”, Line 381:” charge-transfer”, Line 384:” large-quantities”, Line 385:” analytical-techniques”, Line 395:” pharmaco-dynamic”, Line 398:” drug-analysis”.

Comments on the Quality of English Language

Extensive editing of English language required.

Author Response

attached file
